# Multi-Annual Evaluation of Time Series of Sentinel-1 Interferometric Coherence as a Tool for Crop Monitoring

**DOI:** 10.3390/s23041833

**Published:** 2023-02-07

**Authors:** Arturo Villarroya-Carpio, Juan M. Lopez-Sanchez

**Affiliations:** Institute for Computer Research, University of Alicante, 03080 Alicante, Spain

**Keywords:** crop monitoring, vegetation index, Sentinel-1, synthetic aperture radar (SAR), interferometry, coherence, Sentinel-2, NDVI

## Abstract

Interferometric coherence from SAR data is a tool used in a variety of Earth observation applications. In the context of crop monitoring, vegetation indices are commonly used to describe crop dynamics. The most frequently used vegetation indices based on radar data are constructed using the backscattered intensity at different polarimetric channels. As coherence is sensitive to the changes in the scene caused by vegetation and its evolution, it may potentially be used as an alternative tool in this context. The objective of this work is to evaluate the potential of using Sentinel-1 interferometric coherence for this purpose. The study area is an agricultural region in Sevilla, Spain, mainly covered by 18 different crops. Time series of different backscatter-based radar vegetation indices and the coherence amplitude for both VV and VH channels from Sentinel-1 were compared to the NDVI derived from Sentinel-2 imagery for a 5-year period, from 2017 to 2021. The correlations between the series were studied both during and outside the growing season of the crops. Additionally, the use of the ratio of the two coherences measured at both polarimetric channels was explored. The results show that the coherence is generally well correlated with the NDVI across all seasons. The ratio between coherences at each channel is a potential alternative to the separate channels when the analysis is not restricted to the growing season of the crop, as its year-long temporal evolution more closely resembles that of the NDVI. Coherence and backscatter can be used as complementary sources of information, as backscatter-based indices describe the evolution of certain crops better than coherence.

## 1. Introduction

The use of remote sensing data from Earth observation satellites offers valuable tools for crop monitoring, as a periodic, global and non-destructive way to obtain information about vegetated areas. Particularly, synthetic aperture radar (SAR) data offer unique advantages, with the capacity to retrieve information irrespective of the presence of clouds covering the scene and the day/night cycle and with sensitivity to structural and dielectric characteristics of the scene. As such, working with SAR imagery has proven useful in crop monitoring for agriculture [1,2].

In this field, a wide variety of vegetation indices (VI) have been defined in order to qualitatively and quantitatively monitor vegetation covers. They are used as indicators of plant growth, as well as for estimation of biophysical parameters. These VI are constructed as combinations of different observables. For instance, VI derived from optical imagery usually incorporate spectral measurements at different bands [3]. One of the most widely used optical VI is the NDVI [4], which is computed as the normalised difference between the red and near-infrared wavelenghts due to the characteristically different responses from vegetation in these regions of the spectrum. On the other hand, VIs defined from radar data usually rely on the backscattered intensity at different polarimetric channels and other parameters derived from polarimetry [2]. Different radar VI are defined based on the use of dual- or full-pol data. Examples of these could be the use of ratios [5,6] and the RVI [7] for quad-pol data, or the DpRVI, defined using the degree of polarisation from the covariance matrix for dual-pol data [8].

The use of radar satellite imagery gives access to another type of information: interferometric data. Interferometry between pairs of SAR images results in phase measurements related to the geometry of observation and properties of the scene [9]. An observable which serves as a measure of the quality of this phase is the interferometric coherence.

Coherence can be expressed as a product of terms representing the different sources of decorrelation [9,10]: geometrical or baseline decorrelation, volume decorrelation, the signal-to-noise ratio (SNR) and the temporal decorrelation. This last term refers to losses in correlation due to changes in the scene. Several factors can cause temporal decorrelation, e.g., changes in water content (either in the soil or the vegetation), human activities related to agricultural processes (ploughing, sowing and harvesting) and changes in vegetation (growth and senescence, wind-induced movement of leaves and branches, etc.).

The use of interferometry has proven to be valuable for land cover classification [11,12,13,14]. Due to its sensitivity to vegetation changes and the dependence of these on the particularities of the growth stages of different crop species, the coherence has been exploited for the purpose of crop-type mapping [15,16,17,18].

Interferometric coherence has also been used for retrieval of biophysical parameters [19,20], as well as for crop height estimation [21,22,23] and determination of harvest dates [24]. However, its usefulness as a tool specifically for crop monitoring has not been explored in depth.

Recent studies have compared coherence time series from Sentinel-1 (S1) to the NDVI and phenological stages of various crops: maize [23]; maize, sunflower and wheat [25] and bengal-gram and tomato [26], concluding that coherence reliably provides information on the crop growth stages. In [27], a wide variety of crops were considered, and the coherence was found to be well correlated to the NDVI time series. In addition, the effects of compensating the systematic bias in the measured coherence was evaluated, as well as separating the different decorrelation sources. It was observed that the bias correction, while resulting in an wider dynamic range of values, does not offer a noticeable increase in the correlation with the NDVI. Regarding the separation of the coherence components, it was seen how extracting the contribution corresponding to the SNR to isolate the temporal decorrelation was not critical in order to obtain good results.

In the case of this work, the objective is to provide a broader quantitative comparison between the coherence time series and the evolution of the crops in a way similar to how it was conducted in [27], in addition to making general comparisons between the curves and the main growth stages of the vegetation.

Furthermore, there have been recent efforts in the creation of databases for series of S1 coherence data, either at regional [28] or global scales [29]. The complexity of SAR data processing and interpretation limits its use by non-experts. The development of modern tools can be helpful to increase the accessibility of S1 coherence and SAR data in general.

The objective of this work is to build upon these recent studies and carry out a broader analysis of how the time series for the coherence amplitude from S1 compares to the curves of the backscattered intensity and NDVI. This is intended as an assessment on their potential for crop monitoring, with emphasis on comparing the results over a time series spanning several years, covering successive seasons on the same test site. Another point of interest is to explore the use of combinations of the coherence for different polarimetric channels, in a way akin to how vegetation indices are frequently defined. Specifically, the curves of coherence difference and ratio are analysed. Therefore, this work constitutes an extension of the results presented in [27]. Its contributions can be summarised as follows:Additional types of vegetation cover are considered, for a total of 18 different crop species.A multi-annual evaluation has been performed, which serves as a more complete validation for previous observations, as well as to reach broader conclusions.Other indices based in the coherence (e.g., the ratio of VV and VH coherence) have been tested.The performance of the time series covering only the growing season and considering the entire year has been compared.

## 2. Materials and Methods

### 2.1. Test Site and Reference Data

The study area is an agricultural region, entitled BXII sector, in Sevilla, Spain (see Figure 1). It is covered by a wide variety of different crops that rotate over the years. Five consecutive years were considered for the study, from 2017 to 2021. With the launch of Sentinel-1B in April 2016, the S1 constellation started to provide a 6-day revisit time over Europe. For this reason, 2017, being the first complete year with this level of coverage, was chosen as the starting point for the time series.

During this period of time, 18 crops, in addition to land left fallow, encompass the majority of the surface used for agricultural purposes in the area (94–98% depending on the year). The study focused on the following crops: alfalfa, barley, carrot, chickpea, cotton, maize, oats, onion, pepper, potato, pumpkin, quinoa, rice, sugar beet, sunflower, sweet potato, tomato and wheat (Table 1).

A crop type map, acquired from the official land parcel identification system, was used to identify the distribution of crops in the area each year. Wind speed and daily precipitation data were obtained from the *Sistema de Información Agroclimática para el Regado* [30]. Additionally, a calendar with time intervals for sowing, growth and harvesting of each crop was used. These intervals take into account that the exact dates for sowing and harvesting vary depending on the farmer and the parcel. These time intervals have been used as a guidance to delimit the growing cycle for each crop type in order to compare the time series for the different satellite products.

### 2.2. Satellite Data

The Sentinel-1 A/B constellation, in interferometric wide swath mode, offers a 6-day revisit time in the time period from 2017 to 2021, except for late December of 2021, due to the end of the S1 B mission caused by operation problems. All available S1 SLC images from orbit 74, at both VV and VH polarisations, have been used: 60–61 images for 2017–2020 and 58 images in the case of 2021. Around 13 images per year were later discarded in order to remove sudden coherence drops associated with rainfall events (Figure 2). The radar data acquisition over the study area using orbit 74 was conducted around 6 p.m. in an ascending flight direction, with a 31.5–33.5° incidence angle. The preference of orbit 74 over the other two available orbits (147 and 154) was based on the results presented by Villarroya-Carpio et al. [27].

As for Sentinel-2 (S2) imagery, the revisit time over the area is 5 days. However, the presence of clouds covering the scene limits the number of images suitable to monitor the evolution of the crops to 33–41, depending on the year (Figure 3). The products used were reflectance images with Level-2 A processing (Bottom of Atmosphere Reflectances), with a 10 m spatial resolution. The mosaicking of tiles 29SQA and 29SQB from orbit 137 was necessary to cover the study area.

### 2.3. Image Processing

The pre-processing of the satellite products was performed using ESA SNAP toolbox (https://step.esa.int/main/toolboxes/snap/, accessed on 6 February 2023) and Python routines, analogously to how it was conducted in [27]. The pre-processing steps for S1 consisted of selection of sub-swath and bursts, refining of the orbit state vectors, radiometric calibration, coregistration, speckle filtering, estimation of interferometric coherence and geocoding. The resulting products were series of images containing the backscattering coefficient ($\sigma_{0}$) and the coherence amplitude (γ) at both VV and VH channels.

The coregistration was performed for each year using the image in the middle of the series as the *primary* or reference image. A boxcar filter of 19 samples in range and 4 samples in azimuth was used for the speckle filtering and coherence estimation. The filter implemented in SNAP does not support even numbers as filter dimensions; therefore, Python was used. For the interferometric coherence estimation, pairs of consecutive images were used, with the one acquired first being the *primary* and the second acquired image the *secondary*. The systematic bias in the coherence estimation [31] was not addressed, as its effect on the correlation between the coherence and the NDVI was already observed to be negligible [27]. For this same reason, there was no specific step focused on removing the thermal noise.

SNAP was also used for the pre-processing of the S2 images, from which the series of NDVI images was obtained. The steps followed were: cloud masking, using the cloud mask products included with the reflectance images, mosaiking of the two tiles; and computation of the NDVI using bands 4 and 8 (near-infrared and red wavelengths, respectively).

### 2.4. Construction of the Time Series

The series of pre-processed images was used to create the time series for both the SAR and optical data. As mentioned in Section 1, changes in water content in the soil or vegetation can cause decorrelation between pairs of images. This effect is noticeable in the resulting time series for the coherence, where there are abrupt changes (drops) in the coherence values that can be easily associated with rainfall events registered in the daily precipitation data available. These changes are more pronounced when the surface has little or no vegetation coverage and do not have an important impact during the growing season of the crop, since coherence values are already low. The most affected dates were removed from the series (Figure 2) before proceeding with the analysis.

Additional series were created for the ratio between the polarimetric channels for both γ and $\sigma_{0}$, as well as other radar VI. The crop type map for each year was used to extract the pixel values corresponding to each of the crop varieties in the scene. The resulting products were series describing the evolution of the NDVI, backscatter and coherence for each crop during every year. For instance, Figure 4 shows the complete time series in the case of pepper and wheat.

The ratio between backscattering channels remains approximately constant outside the period of vegetation growth. The values of $\sigma_{0}$, as well as the ratio, increase as more scatterers appear on the scene during the vegetation growth. The ripple observed in these curves is due to calibration differences between S1 A and B, and it has been reproduced in other works [27,32,33].

The curves for the coherence reflect the expected behaviour as well: high initial and final values corresponding to periods of exposed soil, a decrease in the coherence during the growth of the crop and until the peak vegetative state, where the coherence remains low and a posterior increase matching the maturation and the harvest. The ratio between coherence channels reflects how the growth of the crop reduces the difference between the channels, as the presence of vegetation causes almost total decorrelation at both channels. The reason the coherence does not reach zero is the systematic bias in the estimation of the coherence amplitude [31]. The same effect could be observed in the analogous time series obtained for 2017 [27]. Finally, the NDVI series follows a similar trend, with the curves rising as vegetation grows and descending with the senescence and eventual harvest of the crop.

### 2.5. Correlation between SAR and Optical Time Series

The comparison between the SAR and NDVI time series was carried out through an analysis of the linear correlation between them. The data points chosen for each date are the average values for the pixels corresponding to each of the crop types. In order to temporally align the pairs of time series, the curves for the backscattering and the coherence were interpolated to match the dates with NDVI data. While working in the opposite way would lead to having more data points to compare, this was the best solution to address potential problems and artifacts with the interpolation around time periods with very sparse optical data, mainly during spring and autumn (Figure 3).

The growing season for each crop was delimited with the help of the crop calendar. Comparisons between pairs of time series were established both for these selected time periods (i.e., only during the cultivation cycle) and for the complete series.

## 3. Results and Discussion

The coefficients of determination ($R^{2}$) for the linear fits between the coherence time series and the NDVI are displayed in Figure 5 and Figure 6. In the first case, the results correspond to the correlations between data points restricted to the growing season for each crop, while in the second case the complete time series was considered for the comparison. Analogous results were obtained for the comparison between the backscatter (particularly the ratio between channels, the RVI and the DpRVI) and the NDVI (Figure 7 and Figure 8).

Regarding the correlations over the growing season of the crops, the overall highest results were obtained when comparing the coherence for the VV channel and the NDVI. In general, these correlations remain similar over the years. VH yields comparably good results in the cases of chickpea, cotton, pepper, pumpkin, sugar beet and sunflower and outperforms the VV channel in the case of carrot and onion. Most of these cases consist of crops with sparser vegetation cover. Other than for carrot, VV shows low correlations for alfalfa, barley, oats, potato, rice and wheat.

A prevalent trend observed in the results is that correlations are lower for earlier crops. These include barley, carrot, oats, potato and wheat, as opposed to other classes that mature around late spring (chickpea, onion and quinoa), midyear (maize, pumpkin, tomato and sunflower) or later (cotton, pepper or sweet potato). As can be seen in Figure 3, the coverage of S2 imagery during the early months of the year was generally sparser due to cloud cover, notably in 2018 and 2020. This has a clear impact on the curves for the NDVI, particularly in the characterisation of the growing cycle of early crops, for example in the case of wheat (Figure 4). This is reflected in lower correlations between the S1 and S2 time series for all the earlier crops with the exception of sugar beet, where $R^{2}$ values are high. As the results for unaffected crops show how coherence is highly correlated with NDVI, it could be assumed that even in these cases, the evolution of the coherence is indicative of how the crops develop.

Alfalfa is a special case due to its multi-annual growing cycle. The plants are partially cut during periodic harvests and left to regrow. Additionally, these harvests do not occur simultaneously for every plot, and in this region, they take place from February through November. This leads to very low variability in both the coherence and NDVI time series. Only the ratio between VH and VV shows some correlation with the NDVI for 2021.

Another crop where the peculiarities of the associated agricultural practices have an important impact in the time series is rice. In this case, the fields are flooded before the sowing period starts and remain covered by water for the whole cultivation period. In the NDVI and backscattering time series, this period is reflected by low values, and the subsequent growth of vegetation and increase in vegetation coverage are reflected as an increase in the curves. On the other hand, a surface of water causes a total decorrelation between the pairs of images. The coherence remains low from the moment the field is flooded until the harvest. Consequently, the backscattering series is a more suitable tool to describe the evolution of this crop.

Finally, in the case of barley and oats, they occupy the smallest surface from all the selected crops, with only a few fields amounting to less than 1% of the total surface (Table 1). Consequently, the presence of missclassified fields and mixed pixels has a more noticeable effect on the averaged values for the time series, which is, for instance, the case for 2019.

The ratio of the coherence measured at both channels, VH/VV, is more correlated with the NDVI than the separate channels in the cases of some crops where poorly characterised curves lead to low correlations, such as alfalfa, barley or oats. It also shows high $R^{2}$ values for cotton, pepper, pumpkin, sugar beet and sweet potato. However, it does not appear to offer an overall improvement over the use of VV and VH.

The correlations between time series covering the whole year (Figure 6) reveal a notably higher variability between years in the results. While $R^{2}$ values are generally lower, they remain high where the results were previously good. Another significant effect of working with the full-year series is that the correlations between VH/VV and NDVI increase in certain cases (pumpkin, rice and sunflower) and remain the same in others (maize and sugar beet). This supports the qualitative observations about the similarities between these series described in Section 2.4.

Regarding the correlations between backscatter-based indices and NDVI (Figure 7 and Figure 8), no clear differences are observed among the different radar VIs. Some of the trends observed for the coherence are still applicable to this case. In most cases, the results obtained for different years are consistent. The worst correlations are observed for crops that grow early in the year (barley, carrot, oats, potato or wheat), probably due to discontinuities in the NDVI series. Additionally, the correlations for certain crops with poorly characterised curves (alfalfa, barley or oats) increase when the complete time series is considered.

As mentioned previously, the time series for the backscatter is better suited than the coherence to describe the evolution of rice, with relatively high correlations ($R^{2}=0.6-0.8$). Likewise, both types of radar products provide similarly good results in certain cases (maize, pepper, pumpkin or sweet potato), and the backscatter outperforms the coherence when comparing to the NDVI series for onion, tomato and wheat. Conversely, the results for cotton and sunflower show low values for the coefficient of determination of the linear fits. The reason for this is the temporal misalignment between the pairs of series. As the considered VIs are directly or indirectly derived from the backscattering coefficient at both channels, particularities of its evolution for certain crops may have unexpected effects. For instance, in the case of cotton, both VV and VH rise similarly during the growth of the plant, resulting in changes in the values of the VI only later during the growing season [27].

Overall, both the coherence and backscatter-based VIs are generally well correlated with the NDVI, and the choice between polarisation channels or different indices vary as a function of the crop type. Although the results obtained are worse for early crops, this is considered to be a symptom of the lack of a complete series of S2 images and not a limitation on the capabilities of the interferometric coherence as a descriptor of the evolution of these crop types. In some of these cases, working with the complete time series is preferable but generally results in lower correlations.

## 4. Conclusions

The first goal of this work was to extend to a multi-year period the evaluation of the suitability of interferometric coherence as a tool for crop monitoring. The results show how the coherence amplitude from Sentinel-1 time series is well correlated with commonly used VIs, and these observations continue to be valid over a period of several years. While the VV coherence generally offers the best correlations, VH coherence works comparably well or better for crops characterised by sparse vegetation cover. The series of the coherence ratio VH/VV provides a potential advantage over those for the separate channels when the complete time series (not limited to the growing season of the crop) is considered. This suggests that the coherence ratio VH/VV can be a better descriptor of the evolution of the crops in certain cases.

The capabilities of coherence as a tool to monitor crops have been evaluated through comparison against optical satellite data. Future research could further confirm these results by a direct comparison to biophysical parameters. Working analogously for different locations could help achieve a wider assessment of the value of coherence as a descriptor of the evolution of different crop varieties. It was observed how the climatic conditions of the test site, particularly the yearly distribution of rains, had noticeable and consistent effects in the observations.

Further efforts are necessary in order to promote wider accessibility and use of interferometric products. This could be conducted facilitating data access through the use of databases [28,29], discerning ways to simplify the necessary processing steps [27] or through a better understanding of the potential applications and limitations of the coherence in this context.

## Figures and Tables

**Figure 1 sensors-23-01833-f001:**
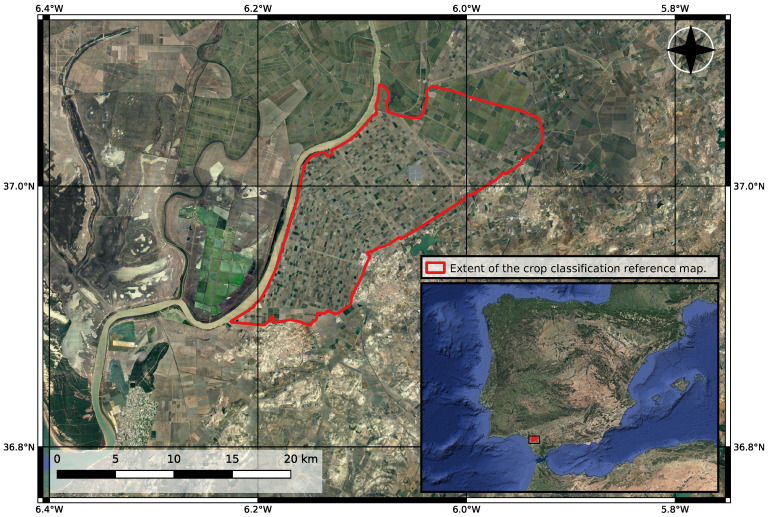
Map of the test site in Sevilla, Spain. The approximate extent of the area covered by the crop type map used for the study is outlined in red.

**Figure 2 sensors-23-01833-f002:**
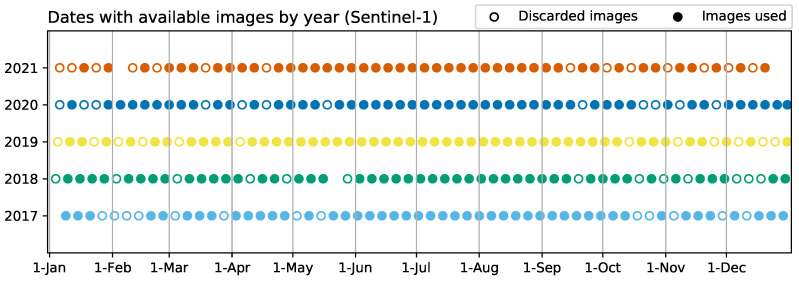
Sentinel-1 images available covering the area during the time of the study. Discarded images correspond to anomalous drops in correlation caused by rain events.

**Figure 3 sensors-23-01833-f003:**
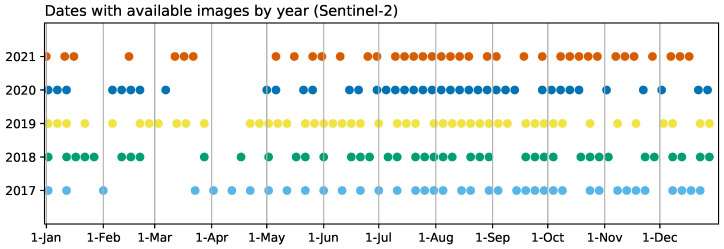
Partial or totally cloud-free Sentinel-2 images available covering the area during the time of the study.

**Figure 4 sensors-23-01833-f004:**
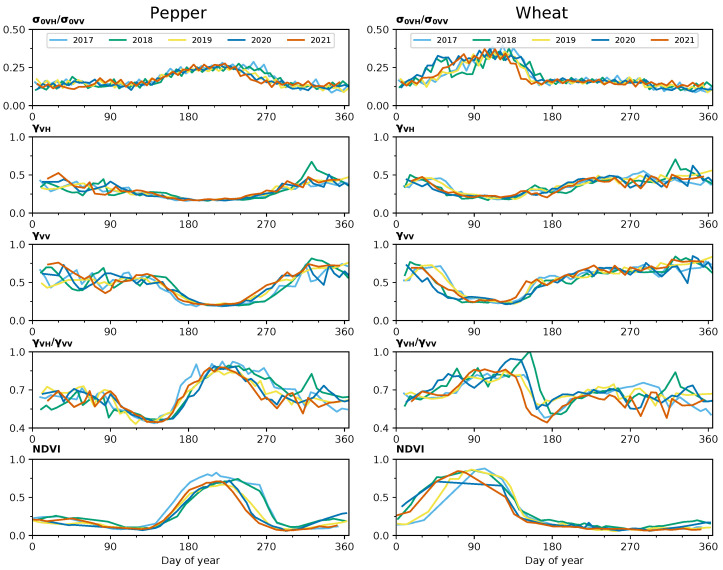
Yearly time series of backscattering, coherence amplitude and NDVI for two of the studied crops. The data points in each curve are the result of the averaging of values for all the pixels corresponding to the crop at each date.

**Figure 5 sensors-23-01833-f005:**
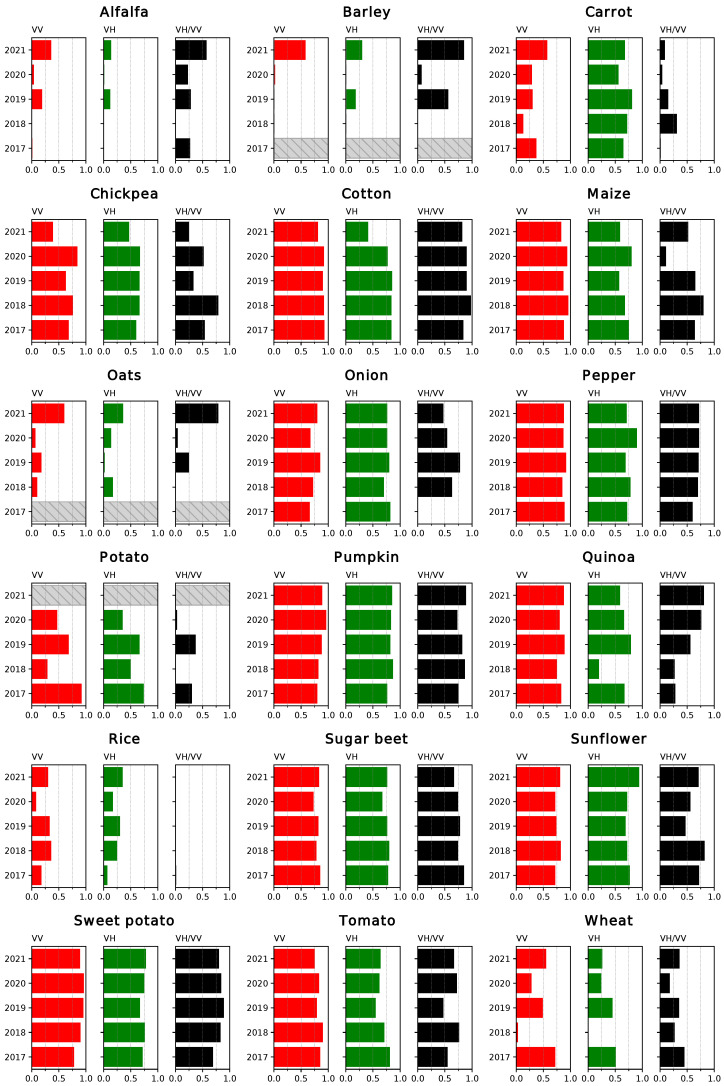
Coefficient of determination ($R^{2}$) for the linear regressions between coherence and NDVI, restricting the series to the growing season. Years without data for a specific crop type are identified with a striped bar.

**Figure 6 sensors-23-01833-f006:**
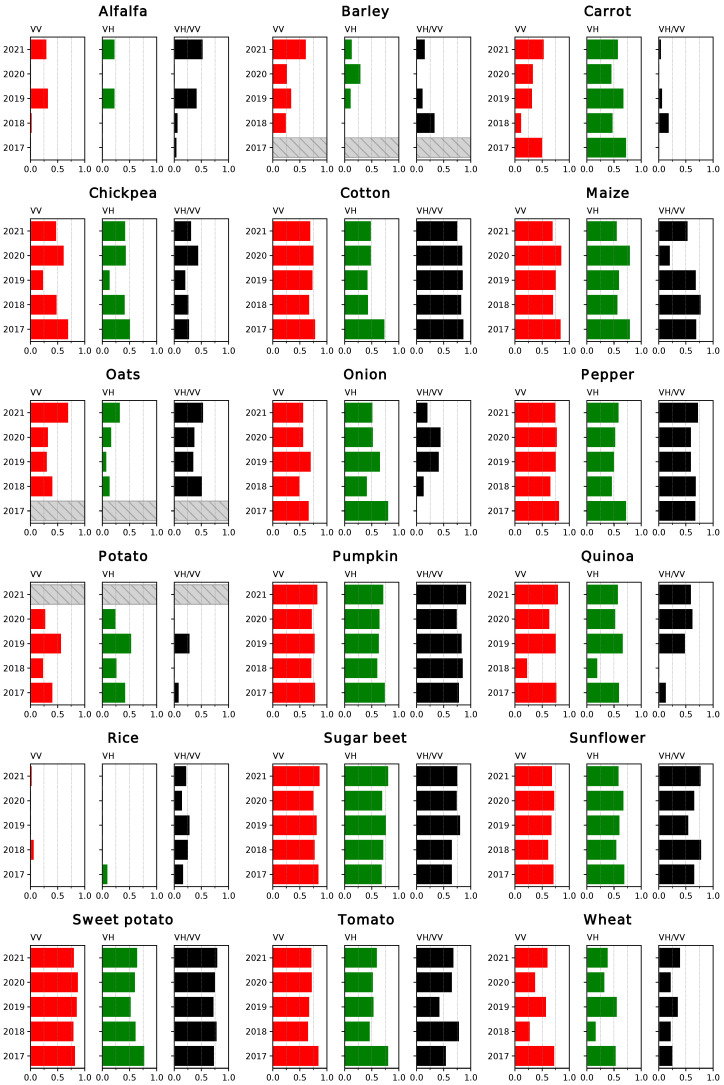
Coefficient of determination ($R^{2}$) for the linear regressions between coherence and NDVI, considering the complete time series. Years without data for a specific crop type are identified with a striped bar.

**Figure 7 sensors-23-01833-f007:**
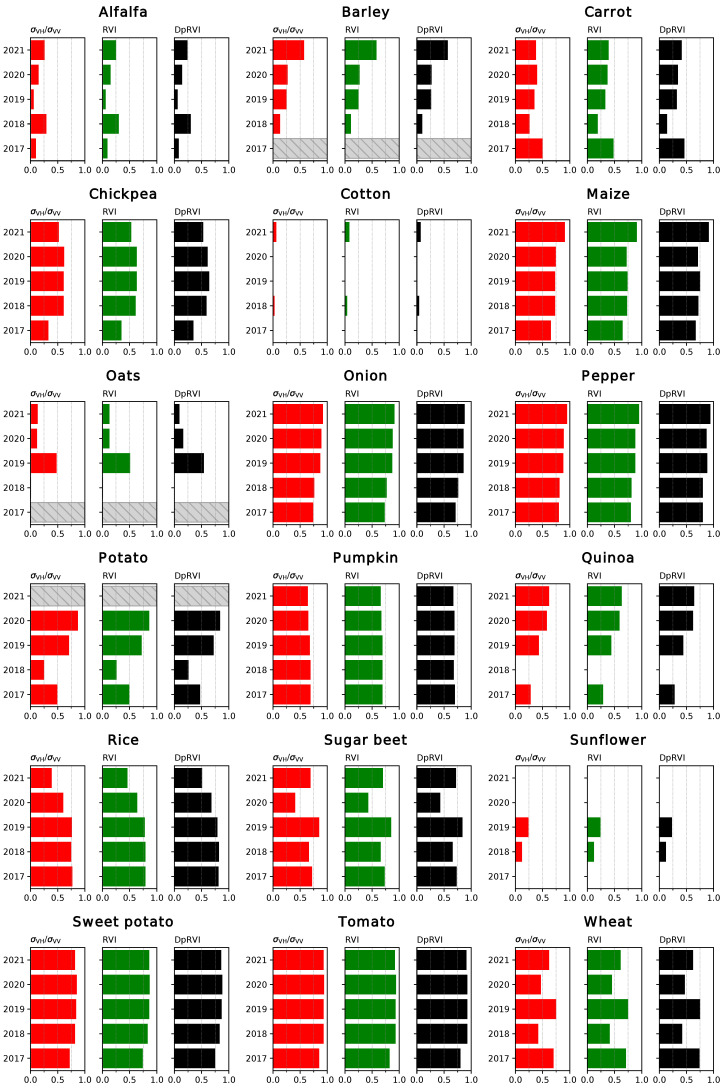
Coefficient of determination ($R^{2}$) for the linear regressions between different radar VI and NDVI, restricting the series to the growing season. Years without data for a specific crop type are identified with a striped bar.

**Figure 8 sensors-23-01833-f008:**
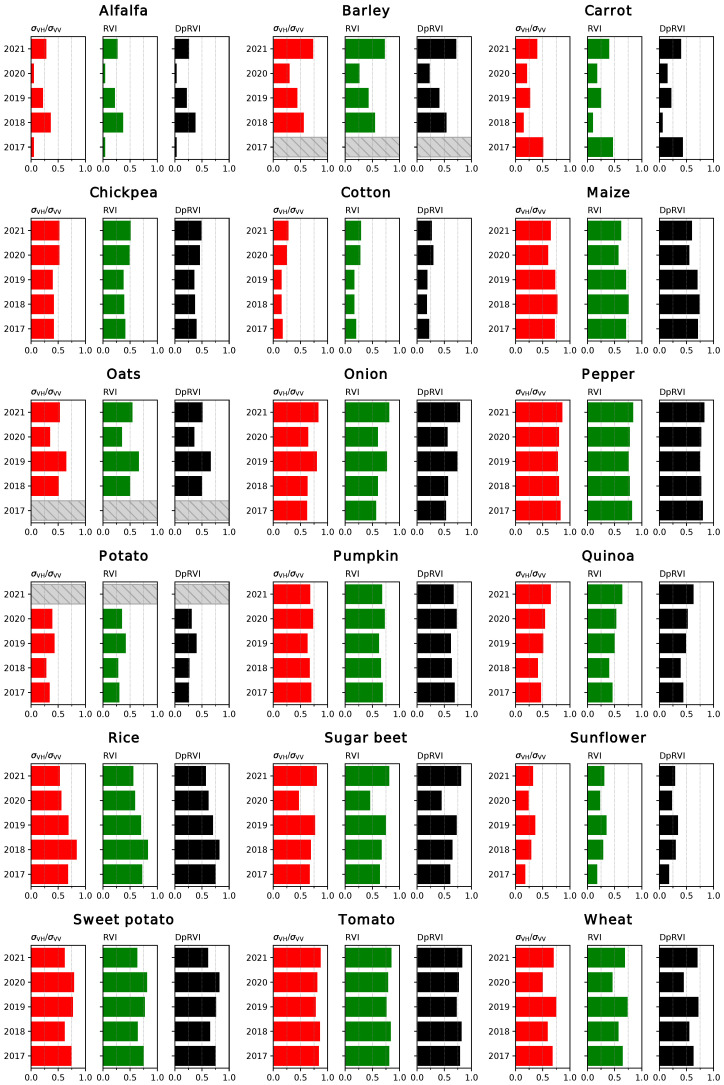
Coefficient of determination ($R^{2}$) for the linear regressions between different radar VI and NDVI, considering the complete time series. Years without data for a specific crop type are identified with a striped bar.

**Table 1 sensors-23-01833-t001:** Total area (km^2^) occupied by each crop every year.

Crop Type	2017	2018	2019	2020	2021
Alfalfa	4.10	3.26	2.14	2.42	3.05
Barley	-	1.81	1.05	0.73	1.06
Carrot	2.22	0.94	1.06	1.82	2.24
Chickpea	0.55	1.02	0.48	1.11	0.61
Cotton	60.47	82.10	78.16	77.69	80.54
Fallow	1.17	4.68	3.36	4.39	19.82
Maize	6.73	5.59	10.81	4.00	4.07
Oats	-	1.69	1.29	0.87	0.18
Onion	1.58	2.25	3.66	6.22	5.36
Pepper	0.67	3.59	4.16	4.68	3.80
Potato	0.43	0.50	0.37	0.88	-
Pumpkin	0.96	1.18	1.17	1.01	0.87
Quinoa	1.08	0.74	1.53	2.31	2.63
Rice	27.81	29.62	30.35	31.19	17.46
Sugar beet	21.23	19.48	14.42	14.81	12.87
Sunflower	6.31	4.79	3.16	4.13	4.32
Sweet potato	1.27	2.04	3.14	2.38	0.56
Tomato	32.46	27.96	35.45	32.88	34.76
Wheat	4.38	4.87	3.29	4.49	2.80
TOTAL	173.43	198.11	199.05	198.02	196.99

## Data Availability

All Sentinel-1 and Sentinel-2 data are open-access and can be downloaded free of charge from different repositories, e.g., Sentinel Science Data Hub (https://scihub.copernicus.eu/, accessed on 30 November 2022), Alaska SAR Facility (https://search.asf.alaska.edu/, accessed on 30 November 2022), etc.

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
