# Peer review of "Multi-Annual Evaluation of Time Series of Sentinel-1 Interferometric Coherence as a Tool for Crop Monitoring"

_sensors, 2023, doi:10.3390/s23041833_

Round 1

Reviewer 1 Report

The article is good work. It can be published if the following operations are performed.

1) Slightly more up-to-date bibliographies should be used in the References section.

2) Introduction part should be developed a little more and examples from current studies should be given.

3) What is the reason for choosing your work between 2017-2021?

4) Compare the results of your study with the results of similar studies. How are your results different from others?

Reviewer 2 Report

The study is useful and interesting. Your paper is well written and has a good structure. Some minor language corrections could be made. Your methodology is clearly presented. The discussion section is very helpful for the readers, explaining the trends observed and providing enough detail and context. Your conclusions are succinct. 

One thought I had reading the paper was that it should be more highlighted that this is work that builds upon your other paper (Arturo Villarroya-Carpio, Juan M. Lopez-Sanchez, Marcus E. Engdahl,Sentinel-1 interferometric coherence as a vegetation index for agriculture,

Remote Sensing of Environment,Volume 280,2022,113208,ISSN 0034-4257,

https://doi.org/10.1016/j.rse.2022.113208) where you already describe in detail the methodology of exploring interferometric coherence data as a tool for agricultural crop monitoring. 

The current study adopts a more extended study period - 5 years instead of one year in the previous study, and looks into a complete time series and the growing season series. Of course, extending the study to a longer time period is very important. I think that the goal of the study should be re-worded, so that this becomes clearer.

The comparison with crop biophysical variables measured in field campaigns and the testing of the methodology in other regions would strengthen the validity and applicability of the study. 
